# Therapeutic Effect of a Newly Isolated Lytic Bacteriophage against Multi-Drug-Resistant *Cutibacterium acnes* Infection in Mice

**DOI:** 10.3390/ijms22137031

**Published:** 2021-06-29

**Authors:** Ho Yin Pekkle Lam, Meng-Jiun Lai, Ting-Yu Chen, Wen-Jui Wu, Shih-Yi Peng, Kai-Chih Chang

**Affiliations:** 1Institute of Medical Sciences, Tzu Chi University, Hualien 970, Taiwan; pekklelavabo@gmail.com; 2Department of Biochemistry, School of Medicine, Tzu Chi University, Hualien 970, Taiwan; 3Department of Laboratory Medicine and Biotechnology, Tzu Chi University, Hualien 970, Taiwan; monjou@mail.tcu.edu.tw (M.-J.L.); 104323103@gms.tcu.edu.tw (T.-Y.C.); w200811@mail.tcu.edu.tw (W.-J.W.); 4Department of Laboratory Medicine, Buddhist Tzu Chi General Hospital, Hualien 970, Taiwan

**Keywords:** *Cutibacterium acnes*, bacteriophages, acne vulgaris, inflammation, apoptosis, phage therapy

## Abstract

Acne vulgaris, which is mostly associated with the colonization of *Cutibacterium acnes* (*C. acnes*), is a common skin inflammatory disease in teenagers. However, over the past few years, the disease has extended beyond childhood to chronically infect approximately 40% of adults. While antibiotics have been used for several decades to treat acne lesions, antibiotic resistance is a growing crisis; thus, finding a new therapeutic target is urgently needed. Studies have shown that phage therapy may be one alternative for treating multi-drug-resistant bacterial infections. In the present study, we successfully isolated a *C. acnes* phage named TCUCAP1 from the skin of healthy volunteers. Morphological analysis revealed that TCUCAP1 belongs to the family *Siphoviridae* with an icosahedral head and a non-contractile tail. Genome analysis found that TCUCAP1 is composed of 29,547 bp with a G+C content of 53.83% and 56 predicted open reading frames (ORFs). The ORFs were associated with phage structure, packing, host lysis, DNA metabolism, and additional functions. Phage treatments applied to mice with multi-drug-resistant (MDR) *C.-acnes*-induced skin inflammation resulted in a significant decrease in inflammatory lesions. In addition, our attempt to formulate the phage into hydroxyethyl cellulose (HEC) cream may provide new antibacterial preparations for human infections. Our results demonstrate that TCUCAP1 displays several features that make it an ideal candidate for the control of *C. acnes* infections.

## 1. Introduction

*Cutibacterium acnes* (*C. acnes*, formerly known as *Propionibacterium acnes*) is an anaerobic, rod-shaped, Gram-positive bacterium that mainly resides on human skin as part of the normal flora. *C. acnes* produces free fatty acids that, once secreted onto the skin, help produce an acidic environment and inhibit pathogenic bacteria [1]. However, it is also an opportunistic pathogen that causes invasive infections [2]. Acne vulgaris, commonly referred to as acne, is a potentially severe skin inflammatory disease associated with colonization of *C. acnes* [3]. About 85% of adolescents are affected by acne vulgaris [4], but it can occur in any age group, with many cases persisting into adulthood [5,6]. Currently, mild-to-moderate acne is treated by topical therapies, including retinoids and antimicrobials such as benzoyl peroxide, erythromycin, and clindamycin [7]. For more severe cases, systematic therapies such as oral tetracycline and doxycycline are usually prescribed [7]. Although antibiotics are effective in reducing the number of lesions, the threat of antibiotic resistance has been increasing. Several studies have shown that the routine and long-term use of antibiotics against *C. acnes* has exacerbated the problem of resistance [8,9,10,11]. The formation of biofilms makes *C. acnes* less susceptible to antibiotic treatment due to reduced penetration of the drugs [12,13]. Because of the emerging situation of antibiotic resistance, seeking an effective alternative therapy is urgently needed.

Bacteriophages (phage therapy) may prove to be a realistic alternative approach [14,15,16,17]. As opposed to antibiotics, bacteriophage is a naturally occurring bacterial virus that specifically targets the bacterial host and does not affect other flora [18]. Phage therapy has also been shown to significantly prevent biofilm formation [19,20]. Whereas some research has highlighted that bacteriophage can also induce biofilm formation, this occurred only under certain conditions [21]. The use of phage therapy to treat bacterial infection in humans has previously been reported in several studies, such as *Staphylococcus*-related eye infection [22], *Shigella*-induced dysentery (shigellosis) [23], *Pseudomonas aeruginosa*-induced otitis [24], and bacteria-related wound ulcers [25]. Phage therapy has also been shown effective in treating multidrug-resistant (MDR) [26,27,28,29], extensively drug-resistant (XDR) [30,31,32], and pandrug-resistant (PDR) [33,34] bacterial infections.

Although studies have reported numerous bacteriophages against *C. acnes* [35,36,37,38,39,40], none to our knowledge have been described in Taiwan or Asia. Since the use of bacteriophages against a bacterium is complicated by high phenotypic and genotypic diversity within populations of both phages and bacteria [41,42], individual strains of a bacterium may be more or less susceptible, or even resistant, to different phages. To achieve a successful phage therapy against *C. acnes*, we isolated and characterized an additional *C. acnes* phage, named TCUCAP1, in this study. A phage-therapy experiment in *C. acnes*-infected mice was performed using the isolated phage. In addition, we formulated the isolated phage into hydroxyethyl cellulose (HEC) cream [36,43] and tested it for lytic capacity and stability under different conditions. 

## 2. Results

### 2.1. Isolation and Characterization of Phage TCUCAP1

The only phage isolated from the skin of healthy volunteers with *C. acnes* as an indicator host was designated as TCUCAP1. Plaques appeared after 48 h incubation at 37 °C, and all plaques were morphologically similar, featuring halos with diameters of 1–3 mm (Figure 1A). Although we saw different sizes of plaques (smaller, medium, and larger), purification by the double-layer agar method still revealed different sizes. Sequencing analysis also suggested that all of these were the same phage. The host ranges of the isolated phages were tested for 18 *C. acnes* strains and one *C. granulosum* (ATCC25746). We found that the TCUCAP1 phages were able to infect all strains of *C. acnes* except *C. granulosum*, suggesting that TCUCAP1 is highly specific to *C. acnes* (Figure 1B). The isolated phage TCUCAP1 was further examined by transmission electron microscopy (TEM) and classified based on the criteria proposed by Ackermann [44]. The results revealed that TCUCAP1 consists of a head with a diameter of 50.4 ± 1.1 nm and a non-contractile tail that is 155.9 ± 31.0 nm long and 9.2 ± 0.5 nm in diameter (Figure 1C, Table 1). Therefore, it is categorized into the family *Siphoviridae*, in the order of *Caudovirales*.

### 2.2. The Phage Genome 

To further understand the characteristics of phage TCUCAP1, we performed genome sequencing using Illumina MiSeq. The TCUCAP1 genome is 29,547 bp with an overall G+C content of 53.83%, 56 predicted ORFs (predicted by Prodigal [45]), and no tRNA genes (Figure 2A, Table 2). Of the 56 predicted ORFs, the average gene length is 398 bp and sizes range from 87 to 2766 nucleotides (Table 2A and Appendix A). However, only 14 ORFs (25%) were predicted with known functions. Forty-two ORFs were assigned to encode hypothetical proteins based on the sequence similarities (Appendix A). The origins of replication regions were not detected by the OriLoc program [46]. Using BLASTn alignments, TCUCAP1 was found to share a high degree of nucleotide sequence similarity with 100 reported *C. acnes* phages (Appendix A) and with sequence identity ranges from 87.86% to 92.78%. The most significant alignment was *Cutibacterium* phage P108C (GenBank: MN813683.1), which has an identity of 92.78% (Figure 2B).

### 2.3. Phage Therapy in Mouse Intradermal Injection Model

To investigate whether phage TCUCAP1 can also prevent *C. acnes* in vivo, 2 × 10^7^ CFU *C. acnes* was injected into both sides of the mice’s backs as previously described [47]. Mice were then given phage therapy in one side of their backs with 2 × 10^7^ PFU of phage and were sacrificed at 24, 48, and 72 h post-treatment (Figure 3A). The results showed that mice injected with *C. acnes* developed inflammatory nodules (Figure 3B,C; Appendix A). On the side of the back of the phage injection, most nodules decreased in size or disappeared, and no nodules increased in size. Similar results were observed in histology. As *C. acnes* injections induced severe inflammation and epidermal hyperplasia, phage therapy significantly reduced these pathological changes (Figure 3D,E). In addition, phage therapy significantly decreased the expression of inflammatory marker IL-1β (Figure 4A,B) and apoptotic marker caspase-3 (Figure 4C,D). 

### 2.4. Lytic Capacity and Stability of the Phage HEC Cream

To investigate the effectiveness of phage activity in HEC cream, we assessed the lytic capacity and stability of different phage HEC cream formulations following storage at different light exposures and temperatures. Assessment of the phage HEC creams showed that 0.5% phage HEC cream retained full lytic capacity after 180 days when stored at 4 °C (light-protected or not) and when stored at 25 °C in the dark. Only storage at 25 °C with light exposure caused a complete loss (100%) of lytic capacity (by 153 days; Figure 5B). One percent phage HEC cream showed sustained lytic capacity at 180 days when it was stored in the dark. Storage with light exposure, at 4 or 25 °C, resulted in the loss of lytic capacity by 70 days (Figure 5C). Two percent phage HEC cream maintained lytic capacity at 4 °C and showed reduced lytic capacity at 25 °C. While light exposure slightly reduced the lytic capacity of 4 °C stored 2% phage HEC cream, storage at 25 °C slightly increased lytic capacity (by around 98 days) compared to the phage stored in the dark (by around 84 days; Figure 5D). 

## 3. Discussion

*C. acnes* is a crucial cause of acne vulgaris and opportunistic infections in hospitalized patients [2]. As antibiotics are becoming increasingly ineffective due to drug resistance, new treatment options are urgently needed [7]. Numerous reports have confirmed that phage therapy is a safe and promising strategy for treating bacterial infections in humans [22,25,29,48]. In the current study, we isolated and characterized a virulent phage, TCUCAP1, that infects *C. acnes*. TCUCAP1 showed unique host ranges of infectivity against 18 strains (100% of all tested strains) of *C. acnes*, including antibiotic-resistant strains. This result corroborates other previously described *C. acnes* phages that demonstrated a high degree of host specificity and a board host range to different *C. acnes* strains [35,38]. TEM analyses indicated that TCUCAP1 resembles a *Siphoviridae* phage, consisting of an icosahedral head and a non-contractile tail. At present, most bacteriophages reported against *C. acnes* have been characterized in the family *Siphoviridae* [35,36,39]. Our findings, therefore, are consistent with other studies. 

The genome of phage TCUCAP1 was sequenced and analyzed. TCUCAP1 was found to share a high degree of nucleotide sequence similarity with other reported *C. acnes* phages (Appendix A). The closest relative phage is *Cutibacterium* phage P108C (GenBank: MN813683.1), which has an identity of 92.78% (Figure 2B). Moreover, both TCUCAP1 and P108C were isolated from the same *C. acnes* host, ATCC6919. Notably, phage P108C was isolated in Los Angeles, CA, USA. Interestingly these two phages were isolated from two distinct regions of the world, since we know that human microbiomes differ significantly across the globe [49]. This result is also in contrast to the diverse phage populations of other bacterial species [33,48,49]. The high genetic similarity of *C. acnes* phages can be explained by their lack of opportunities for lateral gene transfer and recombination: *C. acnes* is an inhabitant of the human pilosebaceous unit (an anaerobic environment characterized by high lipid content), where it lacks other microbes and seemingly their phages [38,50]. Furthermore, the low genetic diversity of *C. acnes* may also contribute to the high genetic similarity of their phages [51].

The products of ORF20 and ORF21 were predicted to be amidase and holin (Appendix A), two proteins involved in the completion of the lytic cycle. With the insertion of holin into the bacterial cytoplasmic membrane, the amidase gains access to and degrades the peptidoglycan, leading to host cell lysis [52]. These data are also similar to previously reported *C. acnes* phages [35,38]. To further our research, we plan to investigate the therapeutic potential of using recombinant amidase or holin to treat *C. acnes* infections, as others have conducted similar work [53,54].

As expected, TCUCAP1 treatment was able to reduce the inflammatory lesions induced by *C. acnes* in mice. The therapeutic effect was found to be time-dependent, as indicated by smaller nodule sizes and fewer inflammatory responses at 42 and 72 h after the phage injection. *C. acnes* infection has also been shown to induce inflammasome activation, resulting in IL-1β secretion [55,56], in addition to activated caspase-3 dependent apoptosis [57,58]. IHC staining suggests reduced inflammation and apoptosis by phage therapy. As noted, inflammation and apoptosis were still detected at 72 h; therefore, future study may require a longer observation time to fully investigate the effect of phage therapy in vivo. More importantly, the *C. acnes* we injected was obtained from a multi-drug-resistant clinical isolate (PS023); our results therefore also suggested the ability of TCUCAP1 to treat antibiotics-resistant *C. acnes*. 

Phage-resistant bacteria were reported in both in vivo and in vitro studies [59]. Because phage selectivity is usually dependent on the cell surface receptor of a bacterial host, receptor mutations may induce phage resistance [60]. Although the rate of resistance to bacteriophage is relatively low, it cannot be overlooked, as natural selection is one mechanism of evolution [61]. Current studies have focused on using phage cocktails or genetically engineered phages to deal with phage-resistant bacteria [60,62,63]. Future studies may also examine additional phage treatments, such as phage–antibiotic synergy or bacteriophage-derived proteins [64,65,66,67]. Notably, whereas antibiotics or phages can kill *C. acnes*, it is their anti-inflammatory properties, not their antimicrobial activities, that provide the beneficial effects for skin inflammation [68]. Therefore, further research into preventing *C. acnes* from over-producing inflammatory toxins, not eliminating them, should be undertaken.

Recent studies suggest that hydroxyethyl cellulose (HEC) gel is safe, able to penetrate the skin, and capable of delivering medicaments to hair follicles [69]. The use of phage-HEC formulations has also been studied [36,70]. We tested the lytic capacity and stability of different phage HEC cream formulations following storage at different light exposures and temperatures. Of all the conditions tested, we concluded that 0.5% phage HEC cream stored at 4 °C in a light-protected bottle results in optimal stability and efficacy of the phage activity. Dressing impregnated with the phage may therefore prove a vital issue for further research, and our future work will concentrate on using the phage-impregnated dressing or gel to treat *C. acnes* infection in vivo and will include both animal testing and clinical trials.

In conclusion, a new *C. acnes* phage named TCUCAP1 was isolated in this study. To the best of our knowledge, this is the first *C. acnes* phage isolated in Asia and Taiwan. This phage morphologically belongs to *Siphoviridae*. The phage showed good therapeutic results in mice, and when the phage was formulated into 0.5% HEC cream, it was capable of killing *C. acnes*, even after 180 days of storage. The phage and its gel may be clinically tested to treat human *C. acnes* infections.

## 4. Materials and Methods

### 4.1. Bacterial Strains and Culture Conditions 

Two reference strains (*Cutibacterium acnes*, ATCC6919 and *Cutibacterium granulosum*, ATCC 25746) of the *Cutibacterium* species were purchased from Bioresource Collection and Research Center (BCRC), Taiwan. One clinical isolate (*C. acnes*, BTCH77596) was collected from the Hualien Tzu Chi Hospital, Taiwan. An additional 16 clinical isolates of multi-drug-resistant *C. acnes* were non-redundantly collected from the National Taiwan University Hospital, Taiwan (Appendix A). All cultures were routinely grown on tryptic soy agar with 5% sheep’s blood (blood agar) and reinforced clostridial medium (RCM; 1.2% (*w*/*v*) agar was added when solid media were required) at 37 °C under an anaerobic atmosphere (10% H_2_, 10% CO_2_, and 80% N_2_).

### 4.2. Isolation and Propagation of Bacteriophage

*C. acnes* bacteriophages were isolated from the skin swabs of healthy individuals. Sampling sites included the forehead, nasal bridge, ala and apex of nose, philtrum, and jaw. This study was approved by the Institutional Review Board (IRB) at the Hualien Tzu Chi Hospital (IRB numbers: IRB106-03-B; date of approval: August 2017 to July 2019), and all donors provided their written informed consent.

The collected swabs were immediately placed into sterile tubes containing 5 mL of phage buffer (10 mM Tris, pH 7.5, 10 mM MgSO_4_, 68.5 mM NaCl, and 1 mM CaCl_2_) and cultured in an anaerobic atmosphere at 37 °C for 48 h. Then, samples were centrifuged at 10,000× *g* for 10 min and the supernatants filtered through 0.22 μm filters to remove residual bacterial cells. To isolate a bacteriophage in the filtrate, a double-layer agar method was performed on *C. acnes* (ATCC6919). Plaques were observed on the plates after 48 h of anaerobic incubation at 37 °C. A single plaque was picked up with a sterile pipette tip and propagated in a new culture with the addition of *C. acnes*. After proliferation, the phages were purified several times using the double-layer agar method. Lastly, purified phages were stored at 4 or −80 °C in 30% (*v*/*v*) glycerol until further analysis.

### 4.3. Host Range Analysis

The host range of the isolated phages was examined by the double-layer agar method on the different *C. acnes* strains listed above. Briefly, 300 μL of *C. acnes* in the log phase, 1 mL phage suspension (10^6^ PFU/mL), and 7 mL 0.7% RCM soft agar medium were mixed and poured onto the 1.2% RCM solid agar plate. The plates were then incubated anaerobically at 37 °C for 48 h. A cleared lysis zone indicated bacterial sensitivity to a phage.

### 4.4. Caesium Chloride (CsCl) Gradient Purification

A high titer of phages (around 10^12^ PFU/mL) was centrifuged at 18,000× *g* for 3 h at 4 °C using a Beckman LE-80K ultracentrifuge (Beckman Coulter, Inc., Palo Alto, CA, USA). The pellet was then resuspended in 1 mL RCM. The resulting phage suspension was placed on a CsCl gradient composed of 1.2, 1.3, 1.4, 1.5, 1.6, and 1.7 g/mL layers and spun at 30,000× *g* for 3 h at 4 °C using a Beckman LE-80K ultracentrifuge.

### 4.5. Transmission Electron Microscopy (TEM) Analysis

CsCl-purified phage particles (10^10^ PFU/mL) were used for transmission electron microscopic analysis. Phages were placed on a freshly prepared formvar-coated grid (300 mesh copper grids), followed by negative staining with 2% uranyl acetate. The grids were observed with a Hitachi H-7500 transmission electron microscope at an acceleration voltage of 80 kV.

### 4.6. Extraction of Bacteriophage DNA

Phage particles were precipitated overnight in PEG 6000 at 4 °C. After centrifugation and resuspension with DEPC-treated water, the phage was mixed with phenol, chloroform, and isoamyl alcohol (29:28:1 by volume). After agitation and centrifugation, the upper layer was discarded and an equal volume of isopropanol was added. Following centrifugation, the pellet was washed, air-dried, and resuspended in DEPC-treated water. The quality of the DNA preparations was verified on an agarose gel, and DNA concentration was determined spectrophotometrically by measuring the absorbance at a wavelength of 260 nm.

### 4.7. Genome Sequencing and Bioinformatic Analysis

Whole-genome sequencing of the phage DNA was performed using the MiSeq system (Illumina, Inc., San Diego, CA, USA). Genomic reads were assembled using Bowtie v1.1.1 [71]. Prediction of all open reading frames (ORFs) was performed by Prodigal [45], and annotation of predicted ORFs was carried out by the Basic Local Alignment Search Tool (BLAST) [72]. The sequence data and annotation information of phage TCUCAP1 were deposited at GenBank under accession number MW505928.

### 4.8. Phage Therapy in Mouse Subcutaneous Injection Model 

Twenty-four six-week-old male BALB/c mice were purchased from the National Laboratory Animal Center, Taipei, Taiwan, and were housed in an animal facility under a 25 ± 2 °C and a 12-h light/dark cycle condition and with free access to water and food. The mice were randomly divided into two groups: A-mice, injected subcutaneously with sterile saline on one side of the back (as a control) and a *C. acnes* suspension on the other side (Appendix A), and B-mice, injected with a *C. acnes* suspension on both sides of the back. Mice in group B were further injected subcutaneously with a phage suspension on one side of the back 24 h after the *C. acnes* injection [47,73]. A *C. acnes* suspension was prepared from an antibiotic-resistant clinical isolate PS023 at concentrations of 10^9^ CFU/mL, whereas the phage suspension was prepared at concentrations of 10^9^ PFU/mL. Both suspensions were subcutaneously injected into the mice’s backs in 20 µL aliquots using a 30 gauge needle. All protocols involving animals were approved by the Institutional Animal Care and Use Committees (IACUC) of Tzu Chi University (No. 105077; date of approval: August 2017 to July 2019).

### 4.9. Examination of Pathological and Histological Changes

Mice were observed for the degree of pathological change at 24, 48, and 72 h after *C. acnes* injection (group A) or phage injection (group B). In addition, three mice were sacrificed and tissue samples were obtained by excisional biopsy at each time marker. Hematoxylin and eosin (H&E) staining was performed as previously described [74]. The slides were examined for inflammation and hyperplasia of the epidermal wall. The degree of pathological changes was classified in an arbitrary score: 0, absent; 1, mild; 2, moderate; and 3, pronounced [47].

### 4.10. Immunohistochemisry Stain

Paraffin sections were deparaffinized then rehydrated with Sub-X (Leica Biosystem Richmond) and 100%, 95%, 75%, and 50% ethanol. The antigens were retrieved by soaking sections in boiling EDTA buffer for 20 min. Next, the sections were treated with 3% H_2_O_2_ and 1% bovine serum albumin (BSA), then the sections were incubated overnight at 4℃ with IL-1β (Abclonal; Cat. #A1112) and Caspase-3 (Abclonal; Cat. #A0214) primary antibody. The sections were then incubated with HRP-conjugated secondary antibody (EMD Millipore) for 30 min and 3,3′-diaminobenzidine (DAB; Thermo Fisher Scientific, Inc., Waltham, MA, USA) for 3 min. Hematoxylin was used as a counterstain for cell nuclei. Sections were rehydrated with increasing concentration of ethanol for 3 min each and finally with Sub-X. The slides were scored based on the cellular staining incidence (%) in each section: 0, none; 1, 1–25%; 2, 26–50%; 3, 51–75%; and 4, 76–100%.

### 4.11. Preparation of Hydroxyethyl Cellulose (HEC) Cream 

HEC cream containing phages was prepared as previously described [43,69]. Briefly, 50, 100, and 200 mg HEC powder (to produce a 0.5%, 1%, and 2% cream, respectively) were weighed and dissolved by stirring into a 10 mL solution mixture containing 9 mL phosphate-buffered saline (PBS) and 1 mL phage solution (10^10^ PFU/mL). The resulting phage HEC cream was then serially diluted with the addition of HEC cream to achieve a final concentration of 1 × 10^9^ PFU/mL.

### 4.12. Lytic Capacity of the Phage HEC Cream

To test whether the cream was capable of lysing host bacteria, an RCM agar with *C. acnes* was prepared, and 10 μL of the 0.5%, 1%, and 2% phage HEC cream were added onto the surface. The plates were incubated at 37 °C for 48 h. A clear zone indicated the lysis of the bacteria.

### 4.13. Stability of the Phage HEC Cream

To investigate the stability of the cream, each concentration of the cream was stored at 4 °C and at room temperature (i.e., 20–25 °C) in light-transmitting or light-protected bottles. The stability of the cream to lyse the underlying bacteria was assessed at weekly intervals up to 180 days. Briefly, 0.01 g of HEC cream was removed at weekly intervals. The sample was completely mixed into the phage buffer (10 mM Tris, pH 7.5, 10 mM MgSO_4_, 68.5 mM NaCl, and 1mM CaCl_2_) to a volume of 100 μL. A 10-fold serial dilution was performed, and 10 μL of these dilutions was placed onto a lawn plate with *C. acnes* (1.8 × 10^8^ CFU/mL). The titers of active phage were examined after anaerobic incubation at 37 °C for 48 h.

## Figures and Tables

**Figure 1 ijms-22-07031-f001:**
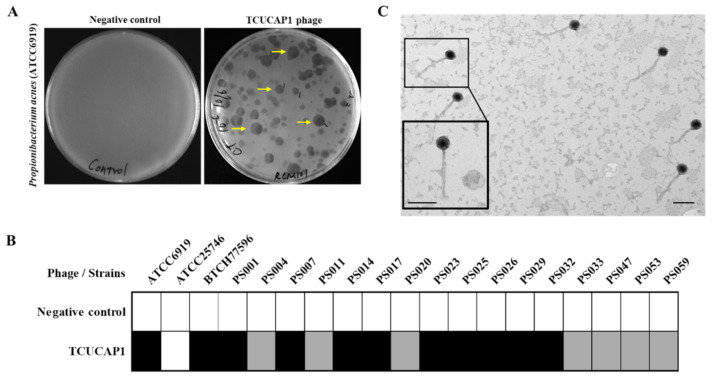
(**A**) Representative images showing spot tests of the isolated *C. acnes* phages TCUCAP1. Yellow arrows showed clear plaques with halos. (**B**) Host range analysis of TCUCAP1. TCUCAP1 was mixed with different bacterial strains. The extent of lysis was observed by eye and is indicated as follows: white cells, no lysis; grey cells, partial lysis (turbid plaque); and black cells, complete lysis (clear plaque). (**C**) Representative electron micrographs showing the morphology of TCUCAP1. The phage was classified as *Siphoviridae* due to their head and non-contractile tail. Scale bar, 100 nm.

**Figure 2 ijms-22-07031-f002:**
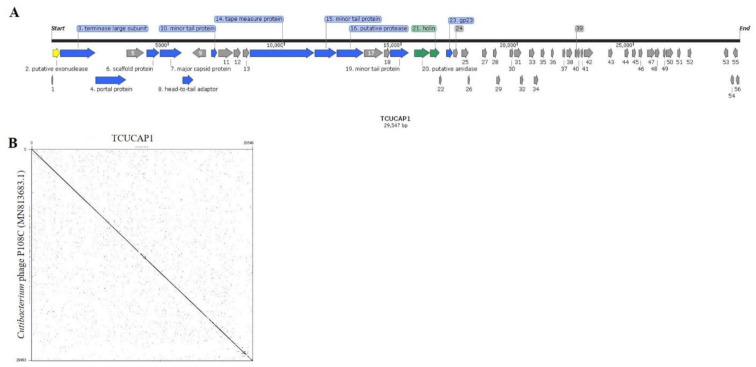
(**A**) Organization of the TCUCAP1 genome. ORFs coding for proteins involved in structure and assembly are marked in blue; ORFs coding for proteins involved in lysogeny are marked in yellow; ORFs coding for enzymes involved in host lysis are marked in green; ORFs coding for hypothetical proteins are marked in grey. Arrows indicate the direction of transcription and translation. The figure was generated using the SnapGene program, http://www.snapgene.com (accessed on 21 April 2021). (**B**) Dot plot nucleotide sequence comparisons of TCUCAP1 and phage P108C were compared using the dot plot program Gepard.

**Figure 3 ijms-22-07031-f003:**
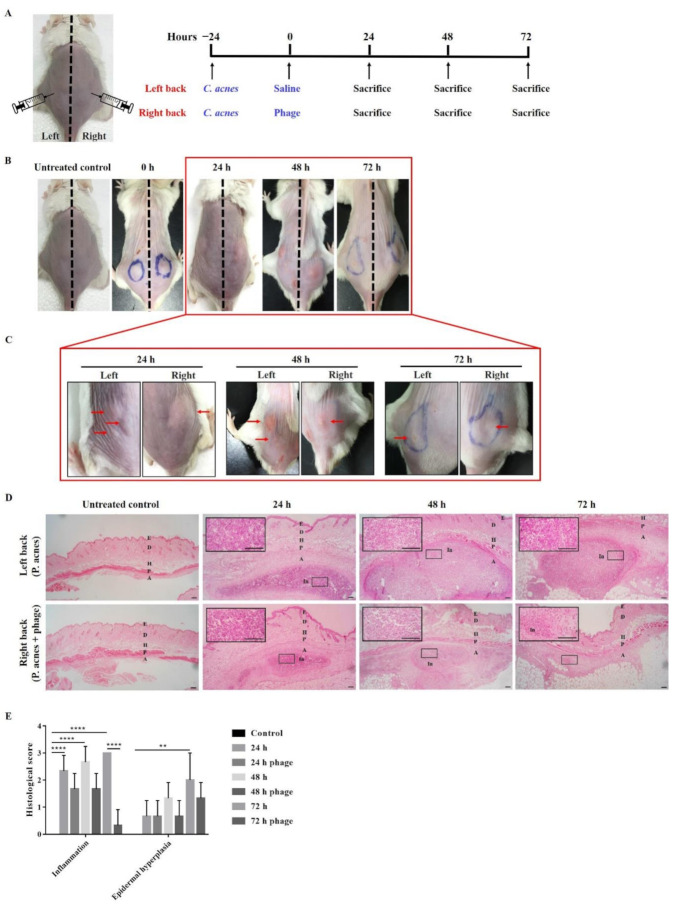
(**A**) Experimental design scheme. We injected 2 × 10^7^ CFU *C. acnes* into both sides of mice backs; 24 h later, 2 × 10^7^ PFU of phage was administered to the right side of the backs. (**B**,**C**) Phenotype of mice injected with *C. acnes* alone or *C. acnes* and phage. After phage injection, the nodule size decreased. (**D**) Representative H&E-stained images showing the histological change in the treated mice (*n* = 3 mice per time point). Images are shown at 40× and 100× magnification with scale bar = 100 µm. E, epidermis; D, dermis; H, hypodermis; P, panniculus carnosus; A, adventitia; In, inflammatory infiltrations. (**E**) Histological score performed on inflammation and epidermal hyperplasia. Results are expressed as the mean ± standard deviation (*n* = 3). ** *p* < 0.01; **** *p* < 0.0001. Significance according to one-way ANOVA with post hoc tests, performed using GraphPrism 6.01 software.

**Figure 4 ijms-22-07031-f004:**
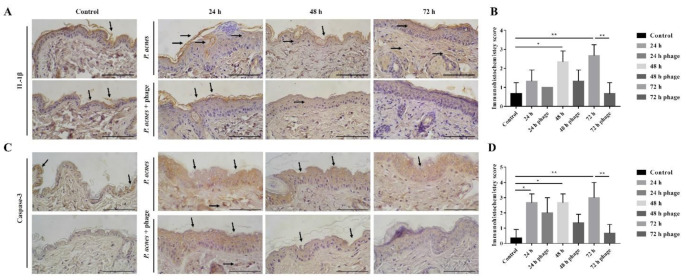
(**A**) Representative images showing IL-1β staining. (**B**) Immunohistochemistry score performed based on staining incidence of IL-1β. (**C**) Representative images showing Caspase-3 staining. (**D**) Immunohistochemistry score performed based on staining incidence of Caspase-3. Arrows show positively stained areas. Results are expressed as the mean ± standard deviation (*n* = 3). * *p* < 0.05; ** *p* < 0.01. Significance according to one-way ANOVA with post hoc tests, performed using GraphPrism 6.01 software.

**Figure 5 ijms-22-07031-f005:**
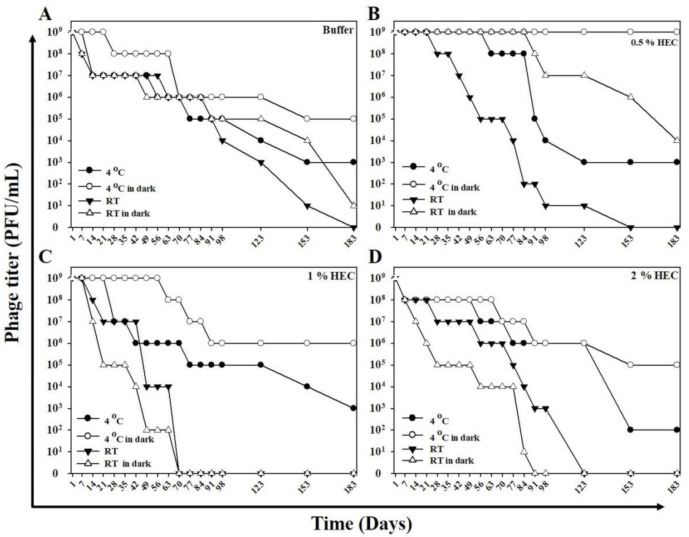
Lytic capacity and stability following storage of the phage HEC creams at different light exposures and temperatures. (**A**) TCUCAP1 in phage buffer as a control group. (**B**) TCUCAP1 per gram of 0.5% cream. (**C**) TCUCAP1 per gram of 1% cream. (**D**) TCUCAP1 per gram of 2% cream. All the cream formulations contained a concentration of 1 × 10^9^ PFU/mL TCUCAP1 phage. ● represents storage at 4 °C in a transparent bottle. ○ represents storage at 4 °C in a light-protected bottle. ▼ represents storage at 22–25 °C in a transparent bottle. Δ represents storage at 22–25 °C in a light-protected bottle.

**Table 1 ijms-22-07031-t001:** Characterization of phage TCUCAP1.

Bacteriophage	Host	CsCl Density (g/cm^3^)	Morphology	Head Diameter (nm)	Head Length (nm)	Tail Diameter (nm)	Tail Length (nm)	Family
TCUCAP1	*C. acnes*	1.4–1.5	Isometric head with non-contractile tail	50.4 ± 1.1	48.5 ± 2.8	9.2 ± 0.5	155.9 ± 31.0	*Siphoviridae*

**Table 2 ijms-22-07031-t002:** General characteristics of the bacteriophage TCUCAP1 genomes.

Characteristic	TCUCAP1
Length (bp)	29,547
Overall G+C content (%)	53.83%
No. of annotated genes	56
Avg gene length (bp)	398.1
Gene density (no. of genes/kb)	1.9
Gene GC content (%)	54.5
No. of tRNAs	0

## Data Availability

The data presented in this study are available on request from the corresponding author.

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
