# Peer review of "Therapeutic Effect of a Newly Isolated Lytic Bacteriophage against Multi-Drug-Resistant *Cutibacterium acnes* Infection in Mice"

_ijms, 2021, doi:10.3390/ijms22137031_

Round 1
Reviewer 1 Report
I loved your article and suggest only minor changes, and those basically in your introduction. C. acnes is immunologically active and has been in the hair follicles of humans for centuries and there are references suggesting that it helps prevent diseases like rabies. Acne treatment is controlling the bacteria and not eliminating them. In short, this bacteria has become part of us. In that case, bacterial resistance is not the parameter of value of treatment, as making C. acnes stop producing inflammatory producing toxins would be therapeutically a success. This little twist in analysis is important in my mind even with bacteriophage therapy.
Reviewer 2 Report
Summary: Lam et al. describe the isolation, characterization, and therapeutic efficacy of a novel C. acnes phage for reducing skin inflammation using a mouse model. Although many phages have been isolated targeting C. acnes, this is the first phage isolated from Taiwan. The paper is well organized with high quality data. Therefore, I recommend publication after minor revisions.
General Comments:
- The authors mention that this phage is the first to be isolated from Asia. The authors should expand their discussion comparing their phage to other phages discovered. This discussion should be expanded beyond the bioinformatics discussion already present. For example, phage TCUCAP1 seems to have broad host range (~Line 75). Is this similar to the other phages isolated?
- Line 47, the authors discuss that biofilm are problematic for antibiotics. However, this is a challenge for phage therapy as well. I suggest the authors expand upon this point in the discussion when discussing phage resistance. Can include the references below.
- Annu Rev Virol. 2018; 5(1):453-476.
- PNAS July 3, 2007 104 no. 27 11197-11202
- Line 70, The Authors should discuss why this particular phage was isolated and chosen for detailed characterization (i.e., Was it the only phage present, or did it produce the largest plaques?).
- Line 73, are these bacterial strains sequenced. They should be sequenced to verify that they are indeed different.
- Figure 1A, there seems to be a range of morphologies, smaller, medium, and larger plaques. Are these the same phage or is there a mixture of phages? Could the authors please describe how the phages were isolated in more detail. I suggest amplifying small, medium, and large plaque lysates and then sequence.
- Figure 1 caption, how is partial lysis versus complete lysis determined? By eye? Please include in text.
- Line 96, add how ORFs were predicted: ‘using Prodigal’
- Throughout the text, claims are made that there is a significant reduction or significant improvement for phage therapy. Please include quantitative values for how much +/- error. For example, Line 116, how much did phage therapy decrease inflammation?
- Figure 3E, are error bars correct? They all appear to be the same.
- Line 127, include how many mice.
- Line 143, by how much was the activity reduced
- HEC cream discussion seems a little out of place since the in vivo data doesn’t use HEC cream for treatment, but I do find these studies interesting and important.
- Figure 5, please make font for the legend and axis larger. Very difficult to read.
- A major claim of the paper is that phage TCUCAP1 is the first isolated from Taiwan. Where was the closest relative phage (P108C) isolated from (Line 176). Why do the authors think the phages are so similar, yet isolated from a different region of the world? Aren’t people’s biomes different depending on the region they are from.
- Line 184, it isn’t clear why these two genes are discussed. Is it because these are the only genes that are different when comparing to P108C. If not, please include this discussion describing the differences between the two phages.
- Please move figure S2 to main text. However, in its current state, it is illegible.
- Line 196, Future studies could also include looking into additional phage treatments.
- Line 200-202, Discussion on phage resistances need expanding. Please include citation listed below:
- 2019, 179, 459–469
